# Revolutionizing Prosthetic Design with Auxetic Metamaterials and Structures: A Review of Mechanical Properties and Limitations

**DOI:** 10.3390/mi14061165

**Published:** 2023-05-31

**Authors:** Muhammad Faris Fardan, Bhre Wangsa Lenggana, U Ubaidillah, Seung-Bok Choi, Didik Djoko Susilo, Sohaib Zia Khan

**Affiliations:** 1Department of Mechanical Engineering, Faculty of Engineering, Universitas Sebelas Maret, Surakarta 57126, Jawa Tengah, Indonesia; farisfardan21@student.uns.ac.id (M.F.F.); djoksus@gmail.com (D.D.S.); 2PT. Bengawan Teknologi Terpadu, Km. 6.5, Wonorejo, Gondangrejo, Karanganyar 65132, Jawa Tengah, Indonesia; 3Mechanical Engineering Department, Faculty of Engineering, Islamic University of Madinah, Al Madinah Al Munawwarah 42351, Saudi Arabia; szkhan@iu.edu.sa; 4Department of Mechanical Engineering, Industrial University of Ho Chi Minh City (IUH), Ho Chi Minh City 70000, Vietnam; 5Department of Mechanical Engineering, The State University of New York at Korea (SUNY Korea), Incheon 21985, Republic of Korea

**Keywords:** prosthetic devices, metamaterials, auxetic structure, negative Poisson’s ratio, metastructure

## Abstract

Prosthetics have come a long way since their inception, and recent advancements in materials science have enabled the development of prosthetic devices with improved functionality and comfort. One promising area of research is the use of auxetic metamaterials in prosthetics. Auxetic materials have a negative Poisson’s ratio, which means that they expand laterally when stretched, unlike conventional materials, which contract laterally. This unique property allows for the creation of prosthetic devices that can better conform to the contours of the human body and provide a more natural feel. In this review article, we provide an overview of the current state of the art in the development of prosthetics using auxetic metamaterials. We discuss the mechanical properties of these materials, including their negative Poisson’s ratio and other properties that make them suitable for use in prosthetic devices. We also explore the limitations that currently exist in implementing these materials in prosthetic devices, including challenges in manufacturing and cost. Despite these challenges, the future prospects for the development of prosthetic devices using auxetic metamaterials are promising. Continued research and development in this field could lead to the creation of more comfortable, functional, and natural-feeling prosthetic devices. Overall, the use of auxetic metamaterials in prosthetics represents a promising area of research with the potential to improve the lives of millions of people around the world who rely on prosthetic devices.

## 1. Introduction

A prosthetic is an artificial device to replace missing or lost limbs; these instances are mainly contributed to amputation [1]. Amputation is a surgical process to remove a segment of a body part. Infection, disease (related to the circulatory system), and traumatic events (falling, being crushed or blasted). Different cases of amputation require different types of prosthetic device based on the level of amputation. In general, prosthetics can be classified into upper limb prosthetics and lower limb prosthetics. Upper limb prosthetics are designed to replace bodily parts above the hip (i.e., trans-radial and elbow disarticulation). On the contrary, lower limb prosthetics are designed to replace bodily parts below the hip (i.e., transtibial, transfemoral) [2]. In prosthetic development, aesthetics and function are two parameters that cannot be separated. Therefore, the development of prosthetics has been extensively studied to provide a better solution for the amputee. The latest development in the field of prosthetics provides not only a primary function but, to some degree, a return for the amputee to the lifestyle they were used to before amputation [3].

Egyptian civilization is known as the first pioneer of developing prosthetic devices, dating from 2750 to 2625 AD. Their prosthetics are mainly made of fibre and used primarily for appearance rather than function; an exception to this is a prosthetic finger found on an Egyptian mummy that is believed to provide a functional and non-functional purpose [4]. Another instance in our history regarding the use or development of prosthetics is from the Punic War (218–210 AD), where a Roman soldier used an iron hand to replace his missing arm [5]. The year 1871 saw a significant increase in amputation numbers following the conclusion of the American Civil War. James Edward Hanger developed an above-knee prosthetic (patented as a “Hanger limb”) with the implementation of a rubber bumper to replace the standard catgut tendons and designed with a hinge at the knee and ankle [6]. Later major wars, such as WW1 and WW2, did not see a significant advancement of prosthetic technology caused by the higher demands for the development of military technology. Later on, to solve this problem, in February 1945, the US National Academy of Sciences (NAS) initiated and planned a research and development (R&D) program aimed at developing prosthetic improvements by applying technology from other specialized fields. This event was soon followed by significant developments such as the first hydraulic knee in 1947, the SACH (solid ankle cushion heel) introduced by Anthony Staros in 1957, a computer-aided robotic arm in 1980, and a motor-powered lower limb prosthetic in 2012 [7]. Subsequently, researchers aimed to implement the unique properties of the MR (magnetorheological) effect as a brake for lower limb prosthetics [8]; by 2020, Brown et al. had conducted research to improve the comfort of amputees by proposing the use of metamaterials as a liner in the socket for a transtibial prosthetic [9]. The diagram within Figure 1 illustrates the development of prosthetics.

Metamaterial is derived from the Greek word “meta” which means “to be beyond”. Hence, a metamaterial is defined as a unique material with effective properties that are beyond its conventional counterparts. The effective properties of a metamaterial originate from its bulk material properties and structural arrangement or design. These effective properties are varied based on the class of the metamaterial and its purpose. Mechanical, acoustic [10], electromagnetic [11], optical [12,13], and thermal [14] metamaterials are the five general classes of metamaterial. Mechanical metamaterials, also known as auxetic metamaterials, are metamaterials exhibiting the unique property of a negative Poisson’s ratio (NPR) or the ability to laterally enlarge when longitudinally stretched. Lakes first studied this unique property in 1987 [15], which was then coined as “auxetic” to shorten the term “having a negative Poisson’s ratio” by Evans et al. [16]. The word auxetic is also derived from the Greek meaning “growing” or “having the tendency to grow.” Since then, the concept of auxetic metamaterial has been extensively researched and is increasing even now.

Early on, the research was mostly focused on developing a better structural arrangement and analysing its properties (i.e., Poisson’s ratio, effective modulus, stiffness, and energy absorption). In terms of its structural arrangement, unique geometries are based on a re-entrant hexagon [17], double-V [18], double-U [19], chiral [20], and even major changes to the already established re-entrant hexagon by adding an additional horizontal component between the vertical parts [21]. Lately, the focus of research in the field of auxetics has gradually shifted to a more practical approach. These research works include an auxetic nail [22], an anti-blast and anti-impact protection [23], an auxetic stent for circulatory disease [24], an auxetic implant [25], and an improvement in foot prosthetic performance [26].

In this review, we provide recent state-of-the-art research in the field of prosthetics and discuss how auxetic metamaterials could provide a more expansive improvement for later research. First, this review will provide the general concept of auxetic metamaterials and their unique NPR property. Then, this review will also report on the latest research and development on these materials in prosthetics, accompanied by a comparative discussion about their advantages, disadvantages, and limitations. Lastly, in this review, we will discuss the future prospects of auxetic implementation for a prosthetic.

## 2. State of the Art

The research of metamaterials’ development has been gradually shifting. The earlier research focused mainly on developing and understanding the relevant mechanical properties of novel metamaterial shapes and geometries. The number of metamaterial research studies in practical and applicative approaches has increased in the last three years. This condition also applies to the practical research of prosthetic devices. In this section, we will provide an overview of the state-of-the-art research with the implementation of auxetic metamaterials as its primary focus. Figure 2 illustrates the state-of-art methods described in this section.

Existing research mainly covers the use and application of lower limb prosthetics due to its high and increasing demands [27]. A common problem concerning lower limb prosthetics usually relates to the comfort of the user [28,29,30]. There have been efforts to solve this problem, including research investigating the use of an auxetic metamaterial inlay to improve the stress distribution associated with the residual limb for transtibial prosthetics [9]. The metamaterial structure implemented in this instance is based on orthogonally patterned equal-sized triangles repeated four times spanning 180 degrees with uniform gaps. The structures are additively manufactured using a 3D printer and “TangoPlus” as the base material. This research analytically investigates the aspects of PS (peak stress) and PPG (peak pressure gradient) with the structure’s wall draft angle variation using the finite element analysis (FEA) method. The comparison and optimizations of each result are discussed to provide a better pressure distribution using a metamaterial inlay. In conclusion, this research is an example of metamaterial implementation in transtibial prosthetics to improve comfort by reducing PS and PPG. Figure 3 shows the auxetic structure, the inlay design and the results of the finite element analysis.

In order to provide an accurate representation of material behaviour, a material model is needed. For this case, the authors have chosen to use the Yeoh third-order representation [31]. Yeoh third-order representation describes the behaviour of materials with a large deformation under loading. C_i0_ defines the three material coefficients of each material base composite material, while σ and ε define the stress and strain produced, respectively. The mathematical model for this is as follows:(1)$$\sigma =\sum_{i=1}^{3}2{\mathrm{iC}}_{i0}\left[\left(1+\varepsilon \right){\left(1+\varepsilon \right)}^{-2}\right]{\left[{\left(1+\varepsilon \right)}^{2}+{\left(2\left(1+\varepsilon \right)\right)}^{-1}-3\right]}^{i-1}$$

Another research work presents a solution for transfemoral amputees by implementing metamaterial in the socket between the inner lining and outer shell [32]. This research also considers the socket’s deformability to improve comfort for amputees by ensuring a sure fit with the residual limb. For this application, a basic re-entrant hexagon honeycomb is designed radially to outline the prosthetic socket’s inner lining. As a purely FEA-based research study, there is no mention of fabrication methods in this article. The material is modelled after the properties of polyurethane for the inner lining part; the outer shell and metamaterial structure are modelled as carbon fibre-reinforced nylon; and the socket dimension was based on the average size of human thigh data.

The solution for the FEM problem was based on Navier’s equation of motion. The mathematical model for the problem is written as follows:(2)$$\rho \frac{\partial^{2}u}{\partial t^{2}}-\nabla \cdot S=0$$

The FEA tool used for this investigation was Solidworks. The results are divided into four investigations. The first two relate to the capability of the socket in supporting a given load. The first was for the static condition of loading the full body weight of the user (standing on one leg) which was assumed to be 70 kg (700 N). The second was focused on investigating the performance of the socket while the user is in gait. This condition was modelled by applying a given impact force during gait (for a 70 kg user, the impact force was assumed to be 980 N (1.4× body weight)).

The analysis includes a force acting from the residual limb toward the inner circumferential surface of the inner lining of the model. This type of load caused a stretching deformation to the inner lining. Although for 70 kg user, the circumferential force acting would be considerably lighter than 700 N, this number could also qualify as a form of safety factor and also provide a better visualization for the effect of swelling. For the last simulation case, an external force acting on the surface of the outer shell was modelled and investigated. This case was used to simulate a condition in which the user chooses to push in the outer shell when wearing a slightly loose socket after inserting the residual limb. Based on the average data, the force exerted from one hand is equal to 250 N; with the assumption that the user pushes with both hands, the applied force becomes 500 N. As a result, this research shows that the socket fulfils its strength requirements while also providing additional adjustment to the changing size of the residual limb through the auxetic effects. Figure 4 below illustrate the interface of transfemoral prosthetic being developed along with the results of finite analysis.

Another way to develop a prosthetic is by improving energy absorption. Such is the aim of research conducted in 2021 to give a foot prosthetic a unique behaviour in the heel part of the foot [26]. This improvement is aimed at providing a better energy absorption during the heel strike. This research provides an example of yet another practical research study for a prosthetic device that uses the basic re-entrant hexagon as the metamaterial structure. Onyx was used as the base material for the model simulation. The proposed design was based on re-entrant hexagon geometry, which is modelled on the heel part of the foot. The variation for the design parameter is based on the relative density (RD) of the overall structure applied. RD itself denotes the ratio of all of the lattice structure (A) to the area of the unit cell (A*s*), written as follows:(3)$$\frac{A_{s}}{A}=\frac{t\left(h+2l\right)}{2l\cos\theta \left(h+l\sin\theta \right)}$$

In order to better investigate the energy absorption capabilities of the proposed design, a comparison was performed to obtain the variation in three different cases of RD: 0.46, 0.23, and 0.12, respectively, for Cases 1–3. The foot model was designed as a 2D surface model and meshed with CPE4R elements. The modelled foot dimension was based on the dimension of a custom-built prosthetic from a past study (AMPRO II). The total length of the foot was 250 mm and the height of the toe part was 63 mm, while the heel part was 125 mm. The force-integral method was used to define the energy absorption (W) capabilities of each case mentioned. This method works by integrating the ground force reaction (GRF) denoted as F, over the deformation (S). The force-integral method can then be written based on the work of [26]:(4)$$W=\int F\;dS\;\;\;\;\;\;\;\;\;\;\;\left(t_{0}\leq S\leq t_{e}\right)$$

In conclusion, this research shows the potential to implement auxetic metamaterials for shock-absorbing functions in lower limb prosthetics. Figure 5 shows the content of this research including the auxetic structure being used along with the results for finite element method.

Later on, the same authors conducted another research work focused on providing an additional mechanism to the lower-limb prosthetic [33]. This innovation is located in the toe part of the prosthetic to provide a unique nonlinear stiffness behaviour for the prosthetic. This research is important because of the low number of prosthetic developments targeting an improvement to the toe part, the main focus usually being the mechanism of the heel part [34,35,36]. Similar to earlier research, this study also uses the basic re-entrant geometry as the metamaterial structure and onyx as the modelled material. Compared to the earlier study, the relative density of the structure remains fixed at 0.4 (value based on calculation from Equation (2)). The variations were performed by dividing the shape and the existence of the bending space/bending zone (Figure 6C,D).

For the methodology, the toe angle and toe torque were investigated and optimized to better mimic the natural behaviour of the human toe. The prosthetic is inclined 25° relative to the ground. The ankle part remains constrained in place. A total of 1000 N of force is applied in the direction normal to the inclined part to simulate the weight of an average human male with 100 kg of weight [33]. The results of the proposed design were then compared with the toe-torque of humans during toe-off. The toe torque follows the formula below [37]:(5)T=r · F

A non-dominated sorting genetic algorithm (NSGA-II) was used in this study to optimize the structure of the prosthetic foot. By using MATLAB, the condition was inputted to generate a model. This model was then simulated through ABAQUS. Based on the results provided from ABAQUS, it was inputted back to NSGA-II [38]. This process was repeated until a solution converged. To further validate the performance of the proposed prosthetic, a comparison with human gait data was also conducted. The ground reaction force (GRF), centre of pressure (COP), ankle angle, and ankle torque were gathered. Figure 6 shows the design along with the results compared against human gait data.

The field of lower-limb prosthetics is not the only field of development where an auxetic metamaterial is potentially useful. For instance, another study was conducted developing orthopaedic bone plates (a rehabilitation device used to repair fractured bones) [39]. The use of an auxetic metamaterial for this development is aimed at mitigating stress shielding and limiting the displacement of fractured bones during the healing phase. Basic re-entrant hexagon honeycomb and missing rib structures were analytically investigated using ABAQUS. For the re-entrant structure, an analytical model used to define the Poison’s ratio is as follows [40]:(6)$$v=-\frac{L^{2}\cos\theta \left(\alpha -\cos\theta \right)}{2\alpha^{2}+L^{2}\sin\theta^{2}}$$

From the above model, a Taguchi design of experiments (DOE) method was used to provide the response of the model for given parameters. In this case, the parameters involved are H, L, and θ resulting in Poisson’s ratio as the response. Based on this design, a model was chosen to further study the effect of unit cell quantity towards the value of NPR. For the missing rib structure, the analytical model used to evaluate the Poisson’ ratio is formulated below [41]:(7)$$v=-\tan\beta \tan\left(\alpha -\beta \right)$$

After the structure had been formed (as displayed on Figure 7), it was then used to model the bone plate using SolidWorks. The CAD was configured by modelling based on the properties of stainless steel 316L, with a thickness of 3.6 mm, and width of 13.05 mm, and also featuring 6 screw holes with 16 mm of spacing. In order to optimize the positioning of the auxetic structure on the bone plate, both experimental and finite element methods were used to decide the most optimal position for the auxetic structure. For this analysis, three model variations were investigated (control, re-entrant, and missing rib). Based on the model, direct laser sintering (DMLS) was used for its fabrication process. DMLS itself is a form of 3D printing utilizing laser power to melt and print metal powder layer by layer based on CAD data sliced beforehand [42,43]. The specimens made were then tested based on the ASTM F-32 using INSTRON machine, with a loading rate of 1 mm/min. This experimental test provided load-displacement data. These data were then derived to quantify the bending stiffness (K) of the bone plate. This was calculated from the maximum slope of the load-displacement data. Based on these data, the structural stiffness (or flexural rigidity) of the bone plate is formulated as:(8)$$E\;l=\frac{\left(2h+3a\right){\mathrm{Kh}}^{2}}{12}$$

The mentioned research works comprise state-of-the-art research conducted in the field of auxetic implementation for prosthetic devices. This shows that by implementing auxetic metamaterials, a new and better innovation for prosthetics could be developed. 

## 3. Poisson’s Ratio and Other Mechanical Properties

Siméon Denis Poisson first defined Poisson’s ratio as a measure of the tendency for a material to undergo a deflection perpendicular to the direction of an applied force. The equation which defines Poisson’s ratio is as follows (Equation (1)) [44]:(9)$$v=-\frac{\mathrm{lateral}\;\mathrm{strain}}{\mathrm{axial}\;\mathrm{strain}}$$

Poisson’s ratio of a material is essential and often overlooked as a constant shared by many materials (which are assumed to have a 0.3 Poisson’s ratio). Table 1 provides a list of some common materials and corresponding Poisson’s ratios.

As could be seen from the above list, the most common materials show a positive Poisson’s ratio with few exceptions. This exception either shows zero Poisson’s ratio, as has been found in cork [49,50], or a negative Poisson’s ratio to some degree such as has been found in certain biomaterials such as cat skin [51,52]. Materials exhibiting a negative Poisson’s ratio (NPR) are also known as auxetic. Although the example shown in Table 1 is compromised by naturally occurring materials, artificially made materials known as auxetic metamaterials are the most commonly found material exhibiting auxetic behaviour.

Auxetic metamaterials are made possible through artificially structuring material so that Poisson’s ratio is affected not only by its inherent properties but also by the structural design. Auxetic metamaterials commonly use re-entrant honeycombs geometry, such as re-entrant hexagons [17], and chiral-based honeycombs, such as hexa-chiral [20] and tri-chiral honeycombs [53].

In order to accurately quantify this number, there are certain methods to measure the Poisson’s ratio of a given material experimentally. The most common way to measure this property is by measuring the deformation happening to a material from lateral and axial directions under tensile or compressive testing [54,55]. The FEA method can also be used to measure the Poisson’s ratio of a material with specific geometries, but this method requires the Poisson’s ratio of the bulk materials involved. The FEA method usually simulates the same boundary condition involved for the experimental method. This method is advantageous to measure the Poisson’s ratio of auxetic metamaterials, considering the difficulty of fabricating a specimen or sample to be tested experimentally.

Earlier research on auxetic metamaterials has focused mainly on developing novel structure geometries. In this instance, to validate the performance of the structure, common properties under investigation included the Poisson’s ratio, stress–strain curve, energy absorption, deformation behaviour under loading, and stiffness. Next, we include a discussion regarding the above mechanical properties. The stress–strain curve defines the relationship between the stress and strain of a material or an object. From this curve, one can also envision deformation for any given stress. There are many ways to measure both the stress and strain of a material or object. The method usually consists of experimentally applying a load to a specimen sample and measuring the stress (commonly by the use of a load cell) and strain (typically using an extensometer). The direction of the applied load differs depending on the data to be investigated—various methods may be used, such as tensile [56,57], compression [58], flexural [59], and impact testing [60].

Understanding both stress and strain is essential because they directly affect the values of other mechanical properties such as strength, stiffness, and energy absorption. These properties are also important in the development of an auxetic metamaterial, depending on its development’s purpose and its relation with other properties. Most of the time, an auxetic metamaterial is characterized by high elasticity. This is mainly attributed to unique structure promoting gaps and voids, leading to lower stiffness values [61]. This limits the roles in which an auxetic metamaterial could be effectively used. Applications in aerospace, automotive, structural or construction, and to some degree medical field require materials with a high stiffness.

Great efforts have been made to solve this limitation. Various solutions have been proposed including modifying the auxetic structure’s geometry or design parameters, adding filler to fill in the gaps of void, and combining different materials to enhance the stiffness. These instances show that enhancing auxetic metamaterials’ stiffness is possible. However, there are limitations as to how far pushing the stiffness is viable and the downside of having the possibility to reduce the effect of NPR [62]. Developing and studying auxetic metamaterials must take both limitations and downsides into consideration. Another point of interest when developing or researching auxetic metamaterials is studying how it deforms, also known as deformation behaviour. Understanding the deformation behaviour of an auxetic metamaterial is essential to better predict its performance and how it would fail. The deformation behaviour of an auxetic metamaterial could be investigated by using the FEA method to analyse the deformation. Combined with equivalent experimental testing, a DIC (digital image correlation) method could be effectively executed to compare both results [63]. This comparison will provide a more accurate representation. There are instances when the unique deformation behaviour of auxetic metamaterials makes them potentially useful for energy absorption [64] and impact-resisting applications [65]. Auxetic metamaterials will inwardly contract when compressed. This leads to a higher concentration of materials at the point of contact [66], contributing to an enhanced energy absorption and capacity to resist impact.

All the above properties mentioned are customizable depending on the structure or geometry designed. This is because auxetic metamaterial properties are derived from their inherent material properties but also the geometry of the structure. The auxetic metamaterial’s geometry can be classified into re-entrant-based geometries, chiral-based geometries, and rotation-based polygon geometries. Re-entrant denotes the phenomena of a polygon with vertice(s) pointing inward. The most common re-entrant geometry is the basic re-entrant hexagon also known as the bow-tie hexagon. This type of geometry is able to exhibit a negative Poisson’s ratio behaviour because its re-entrant aspect makes it so that when pulled, the vertice(s) pointing inward is prone to be pulled out when tensile. While in compression, the inward vertice(s) is pushed further in.

The development of the re-entrant hexagon was oriented toward improving stiffness properties. In order to achieve this, an insert material can be used, such as foam. Another approach for the same objective would be to modify the geometry, such as the introduction of additional turning points [67] or additional struts [21]. Double V, also known as double arrowhead (DAH), geometry is—as the name suggests—based on the shape of an arrowhead [68]. This geometry is also considered a re-entrant geometry because there is a single vertice that is pointing inward. Similar deformation behaviour applies when being compressed or tensile. Similar to the re-entrant hexagon, improvement can be made in terms of its stiffness by adding an additional ligament [69]. Double V can also be developed into Double U to reduce the stress concentration made by its vertices [19]. Figure 8 illustrates the development and improvement of re-entrant geometries.

Chirality is a condition in which an object cannot be superimposed on its mirror plane [70]. Recently, chiral metamaterials have been produced made up of lattice geometry that is prone to spin or rotate around its central node [71] when in compression or tensile. As mentioned, a chiral geometry consists of a central node and a connecting ligament. The basic chiral structure is based on the number of connecting ligaments, such as the hexachiral (six ligaments) [20], tetrachiral (five ligaments), and trichiral (three ligaments). A more recent development is focused on the derivation of these basic structures. These instances include the development of an anti-chiral structure, the investigation of the effect of irregularity and disorder [72,73,74], and the modification of the central node or its ligaments [75,76]. Figure 9 illustrates the development and improvement of chiral-based geometries.

Rotation-based polygons differ from the other two lattice classes, which take advantage of thin beam-like parts also known as ligaments to connect the different parts of the structure. On a rotation-based polygon, the connection is made directly on the vertices of the cell. This connection works as a hinge that rotates (opens and closes) the cells based on the force applied [77]. This class is subclassified based on its base geometry, such as rectangular [78,79], triangle [80], and parallelogram [81]. Figure 10 illustrates the different subclasses of the rotation-based polygon.

Early on in this section, it was mentioned that the research and development of auxetic metamaterials has mostly used the FEA method, alongside certain experimental testing with considerable limitations and required adjustments. ABAQUS, LS-DYNA, and ANSYS are commercially available FEA tools that have been used by earlier studies. To obtain a more accurate simulation result, a validation method was used to compare this method with theoretical analysis, experimental testing, or past data from other documentation. Table 2 below lists the tools, base material model, and validation method of a combination of past studies regarding auxetic metamaterials.

## 4. Regarding Prosthetics

As mentioned in the Introduction earlier, a prosthetic is an artificial device to replace missing or lost limbs [102]. These instances are mainly attributed to amputation. Amputation is a process of disarticulating limbs or body parts. Common causes for amputations are diabetes, cancer, traumatic events, hypertension, hyperlipidaemia, and in rare cases include congenital limb deficiencies [103]. Diabetes is a disease caused by a drastic elevation in the blood sugar level in the circulation system. This condition often leads to damage to blood vessels and nerves and impairs the circulatory system. An impaired circulatory system prevents the body from properly circulating blood. An improper circulatory system is particularly dangerous. Should the patient experience a traumatic experience that could lead to an injury, wound, or broken certain body parts, it cannot heal properly. Improperly healed wounds or injuries require immediate treatment as the wound can be a starting point of infection which could quickly spread to other body parts. In order to prevent the spread of infection, disarticulating infected body parts (amputation) is required [104].

Traumatic amputation is an immediate disarticulation of body parts resulting from an accident or injury. Replantation is a surgical procedure to reattach or replant already disarticulated limbs or body parts. There are some factors that need to be taken into consideration to choose between replantation and amputation, this consideration and factors are listed in Figure 11 [105,106,107,108,109].

Amputation level is closely related to the classification of prosthetic devices. Generally, prosthetic devices are classified into two main categories which are lower-extremity and upper-extremity prosthetics. Figure 12 illustrates the classification of prosthetic devices based on the level of amputation [110].

The ratio of lower limb amputation compared with upper limb amputation could reach a ratio of 11:1 [111]. A transtibial prosthetic is divided into an interface, suspension, shank, joint, and foot part. To better visualize the composition of transtibial prosthetics, refer to Figure 13 [102].

An interface is a part of the prosthetic having the role of connecting the residual limb to the prosthetic. An interface must be able to provide fitting, cushion, and shock absorption. The interface part is divided based on the material into hard material (such as wood), and soft material (such as closed-cell foam). Materials such as wood are used early on thanks to their simple manufacturing methods and abundant, easy-to-find materials. In terms of maintenance, this type of interface material is relatively easy to maintain thanks to its high durability. Despite its ease of production, this type of interface material comes with inherent disadvantages, mainly its lack of cushioning. This disadvantage is the reason for later consideration of a soft material interface. Common materials used for soft interface materials include closed-cell foam. This material is used because of its waterproof properties, accompanied by its malleability which makes it a material that is easy to mould based on user characteristics [112]. Suspension refers to the part which holds the prosthetic in place when worn. A good suspension should be able to provide zero relative movements against the residual limb. A badly fitted suspension would cause a condition known as “pistoning”. Pistoning can and does cause pain, skin breakdown, irritation, and non-responsive prosthetic movement.

The choice of the best type of suspension (waist belt, joint and corset, and sleeve) and interface parts for a transtibial prosthetic is different from person to person. Essentially, other than to replace a missing ambulatory capability, a prosthetic is also aimed at the comfort of its user. This means that the best choice lies in what makes the user most comfortable. Whichever type of interface is chosen, there has to be an accurate impression technique to support it. Such impression techniques consist of hand casting, pressure casting, and optical scanning. Apart from the impression, another aspect of fitting a prosthetic involves the act of alignment. The bench alignment method is one of the most common methods of alignment, although there exists a more advanced and accurate alignment technology known as electronic alignment. This method requires the use of sensors fitted into the prosthetic to provide data in terms of user gait behaviour when wearing the prosthetic. This method is capable of providing data that are unavailable from bench alignment, such as the socket load over a period of time walking.

## 5. Limitation to Implement Auxetic Metamaterials

Each study has been generally described in the reference provided in Section 2 regarding the state-of-the-art methods for auxetic implementation in a prosthetic. This section will cover more detail regarding the limitation and comparison of the described research [9,26,32,33,39]. The auxetic single-layer liner studied by Brown et al. was mainly conducted using the FEA method. The material samples were also fabricated so as to mimic the ASTM D575 standard for compressive testing on rubber materials to be used along with the following FEA. The matter of using metamaterials is not considered in said standard; thus, some form of alteration must be made to provide the data required. Alterations included altering the specimen’s thickness to 4.76 mm (from 13 mm) and the loading rate to 4.4 mm/min (from 12 mm/min). In this study, fabrication was only performed for sample specimens. There was no mention of the possibility of full-unit fabrication of the auxetic liner.

Employing another perspective, another study focused on implementing an auxetic metamaterial directly into the inner lining of the socket for a transfemoral prosthetic. In this study, no experimental data were gathered; the properties of the materials used were presented as is. The studies discussed above regarding the implementation of the socket liner both gather data mostly from FEA analysis. Neither of these studies provide any argument as to why no prototype was fabricated to support the proposed design. It must be noted that qualitative data from the user are equally important to better provide comfort for them. The proposed liner for transtibial prosthetics used 3D-printing technology to fabricate sample specimens. However, using the same method to fabricate a fully functioning prototype might be difficult. These difficulties are attributed to the large overall dimensions and the highly complex structure of the metamaterials, which require an equally large and highly accurate 3D printer.

Heon-su Kim et al. proposed the implementation of an auxetic structure for both the heel side and toe side of the foot part. It was mentioned that if the proposed designs were made using a 3D printer, it would come with considerable limitations. Such limitations include a limited weight-bearing capability of the 3D-printed prototype (adjustments are necessary depending on the size and weight of the user) and (b) a 3D-printed foot is limited in terms of providing rapid stiffness change. The last study we will discuss in this article regards the implementation of auxetics for a bone plate. This represents an example of a study capable of fabricating a prototype for testing. It might be attributed to the overall larger unit cell dimension and fewer quantities. This, in turn, eases the 3D-printing process required to fabricate the prototype bone implant. To summarize the discussion in this section, we present an overview of the limitations to ease the comparison of all the state-of-art research in Table 3.

## 6. Future Prospects

In order to better validate our claim, we will also provide particular examples of early development into the implementation individually. Overall, the future prospects can be divided into potential uses in a protective application, robotic application, and aerospace application. The potential for a protective device application results from the unique way it deforms. Past research has shown that an auxetic sandwich panel experienced a local internal contraction in response to force from impact [66]. This will support the sandwich panel to resist damage and absorb more energy from the impact. Studies have shown that compared to conventional sandwich panels and homogenous materials, an auxetic sandwich panel has proven to have an effective potential in this field.

In terms of robotic applications, auxetic material offers the capabilities of form and shape matching and, depending on the material used, is likely to have a higher flexibility. Shape matching has been proven viable in non-robotic studies [86]. A soft auxetic metamaterial has been further explored to be implemented in a soft robotic gripper [113]. Implementing auxetic metamaterials in an inchworm-type soft robotic will ease it to crawl through narrow channels. This is attainable through the high flexibility an auxetic metamaterial provides. The most enticing advantage of auxetic metamaterials in aerospace applications is their lightweight properties. As mentioned earlier in this article, metamaterials mainly consist of lattice or highly porous structures filled with voids. This leads to lighter materials compared to conventional ones with the same overall dimensions. Research has explored the idea of auxetics in aerospace, including the application in a satellite antenna [114]. Another study has also proven the possibility of combining different materials on the same auxetic metamaterial, forming what are known as multi-material auxetic metamaterials [115]. This opens up nearly infinite possibilities to customize each auxetic metamaterial based on specific needs and requirements. Instances of research investigating the performance of multi-material auxetics include those combining the soft properties of FlexPro elastomers and a polyurethane shape memory polymer for its smart properties [91].

Although the research studies provided regarding the developments of prosthetic application have been limited to auxetic metamaterials, as auxetic metamaterials are generally more known, other types of mechanical metamaterial also exist, such as negative stiffness metamaterial [116,117]. There has yet to be any study focused on implementing this for prosthetic design, making it a rich prospect to be explored deeper.

Throughout its history, the development of auxetic geometries has been explored thoroughly. However, novel geometries have been found nearly every year since the concept of auxetic metamaterials first began. From the earlier geometries such as the re-entrant hexagon studied by L.J. Gibson in 1997 to more recent geometry such as the hourglass lattice proposed in 2021 [118], this trend shows that even the generation of newer geometry still has a high potential to be further explored, ranging for more varied base geometries. Lastly, to summarize the content of this section, Figure 14 illustrates the potential and future prospects of auxetic metamaterials that have been discussed above.

## 7. Conclusions

To summarize, the trend of auxetic metamaterials research has been shifting to a more practical subject in the last couple of years. Based on the already completed and ongoing studies on this subject, auxetic metamaterials have the potential to be developed primarily for use in prosthetic devices, protective devices, robotic applications, and aerospace engineering. The above practical focus does not by any means stop any further development in terms of theoretical research, such as the advancement of the metamaterial structure. From another perspective, as has already been proven by other studies, the act of combining different types of materials in a single auxetic structure is proven theoretically feasible in achieving a more complex and customized property. In order to better support the content of this article, this document has also provided some of the most basic information regarding both prosthetic and auxetic metamaterials. For auxetic metamaterials, this includes a description and discussion regarding its properties, such as the negative Poisson’s ratio effect and other relevant properties. On the subject of prosthetics, references are taken from already published pieces of literature to better understand prosthetics generally, in hopes of sparking interest concerning the potential of implementing auxetic metamaterials in prosthetics. In conclusion, this article provides an overview and discussion regarding the research of auxetic metamaterials, primarily for prosthetic devices. The information provided in this article is targeted to highlight not only the positive advantages of using auxetic metamaterials but also the negative disadvantages and limitations. In addition, the later section provides a discussion regarding the analysis of the future prospects regarding auxetic metamaterials based on completed studies conducted on this subject within the last decade. This article may serve as a go-to document for any new researcher interested in developing the technology of auxetic metamaterials further, mainly in terms of its applications for prosthetic devices.

## Figures and Tables

**Figure 1 micromachines-14-01165-f001:**
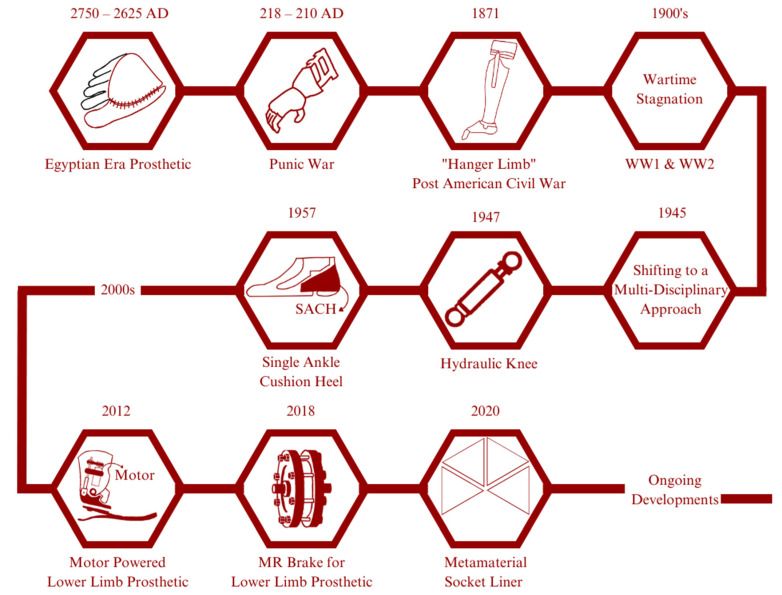
Historical development of prosthetics: In ancient Egypt, wooden prosthetic toes and feet were crafted for amputees. These prosthetics were not functional but were designed to mimic the appearance of a real foot or toe. Similarly, in ancient Greece and Rome, prosthetics were made from wood and bronze and were designed primarily for aesthetic purposes. In recent years, advancements in robotics and computer technology have led to the development of advanced prosthetic limbs that can be controlled using neural signals from the user’s brain. These “bionic” limbs are capable of incredibly life-like movement and have revolutionized the field of prosthetics. The development of prosthetics has been a long and fascinating journey, with significant advancements being made over the centuries. Today, prosthetic technology continues to advance rapidly, offering new hope and possibilities for amputees around the world.

**Figure 2 micromachines-14-01165-f002:**
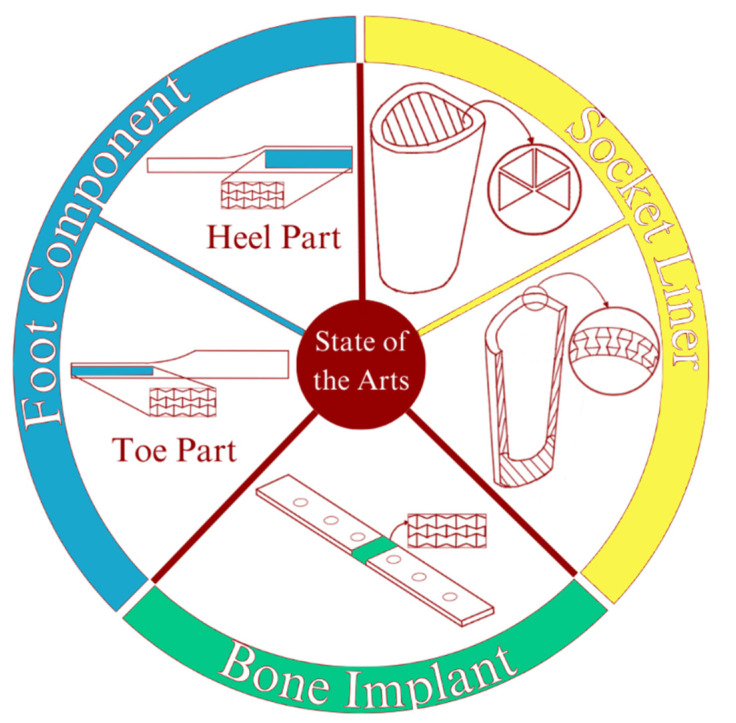
The use of auxetic materials in prosthetics is still in the early stages of development; there is promising research being conducted in this area. As the technology and understanding of auxetic materials continue to evolve, we may see more widespread use of these materials in prosthetic design and development in the future. One of the developments implemented regarding materials is the use of metamaterials. A small amount of research has been conducted on metamaterials to change and enhance the properties of their prosthetics.

**Figure 3 micromachines-14-01165-f003:**
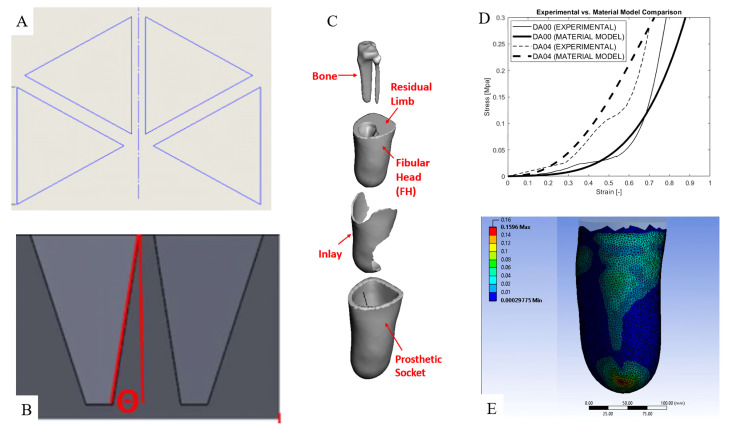
Implementation of auxetic metamaterials for the interface part of a transtibial prosthetic. (**A**) Unit cell of four triangles orthogonally arranged. (**B**) Variation produced on the draft angle of the wall. (**C**) The layout of the inlay in regard to the inlay and residual limb in an exploded view. (**D**) The results of Yeoh third-order representation show good agreement for a higher draft angle when compared with experimental data taken from experimental testing of the material sample. For the lower draft angle, the model shows a slight error **caused** by the inherent limitation for Yeoh third-order representation in representing a drastic change in buckling, such as the condition that exists for the lower draft angle. (**E**) Visual representation for the FEA results to analyse PS and PPG. This resulting gradient reduction capability is compared to the “no inlay” condition [9].

**Figure 4 micromachines-14-01165-f004:**
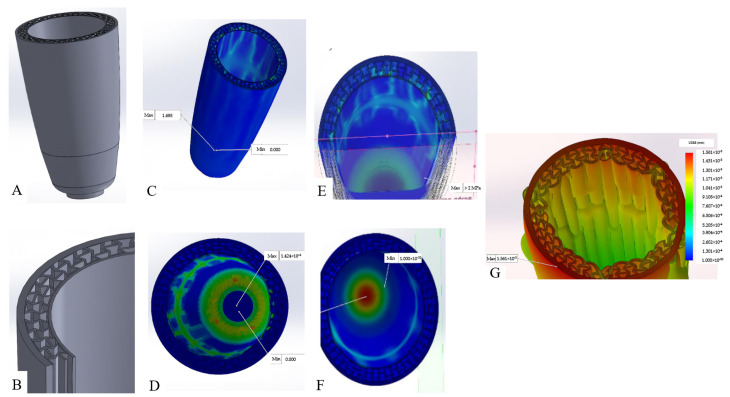
Implementation of auxetic metamaterial for the interface of a transfemoral prosthetic. (**A**) The external view of the interface part, consisting of the outer shell, metamaterial structure, and inner lining. (**B**) The metamaterial structure connecting the inner lining and the outer shell. The structure is modelled after the re-entrant hexagon arranged radially with carbon-fibre-reinforced nylon as the material. (**C**,**D**) The resulting view of the case of the simulation for the user statically standing on one leg. (**C**) The von Misses stress distribution (maximum reduced stress present on the lower part of the socket with 1.695 MPa of stress) of this case, with (**D**) the deformation distribution (maximum deformation for the applied force is 1.42 mm). (**E**,**F**) The results for the case of a walking user. (**E**) The stress distribution (with the maximum value exceeding 2 MPa, which is still lower than the strength of the material modelled) for this case, and (**F**) the displacement distribution (the maximum displacement does not exceed 4 mm). (**G**) Lastly, the result for the case of an external force acting on the outer shell. For this case, the maximum stress occurring is 0.27 MPa [32].

**Figure 5 micromachines-14-01165-f005:**
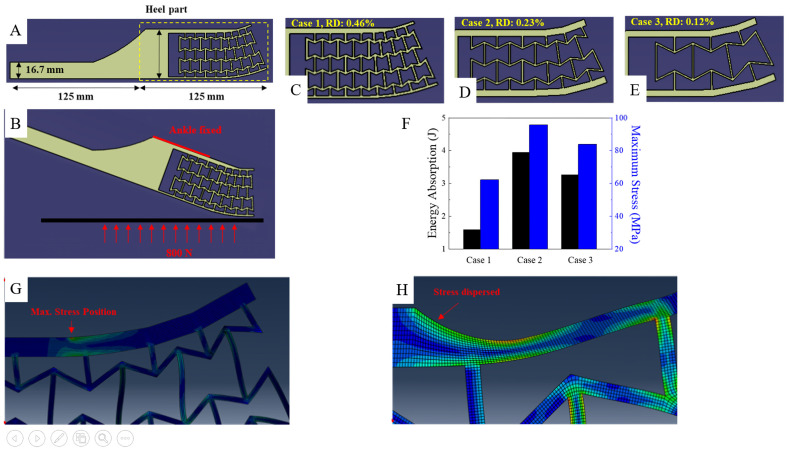
The use of auxetic materials in prosthetics for the heel part of the foot. (**A**) The auxetic metamaterial structure was based on a re-entrant hexagon and designed to be located on the heel part of the foot with the overall dimensions as visualized. (**B**) The boundary and simulation condition are described as the foot positioned a certain degree in relative to the ground, while the ankle equivalent is fixed. (**C**–**E**) The model investigated is divided into three categories based on its RD, with the values of (**C**) 0.46 for Case 1, (**D**) 0.23 for Case 2, and (**E**) 0.12 for Case 3. (**F**) The comparison of simulation results in regard to the energy absorption and maximum stress in each case. From this result, it can be concluded that the case shows the most promising design for absorbing energy; this, however, is also the condition with the highest maximum stress compared to the other two cases. (**G**) The high value of maximum stress was caused by a stress concentration occurring in the middle part of the upper plate. (**H**) To alleviate this problem, a rounded shape was proposed. This solution has proven to be effective as the maximum stress for the rounded shape design has been reduced to 37.92 MPa (within the boundary of onyx yield strength of 36 MPa) [26].

**Figure 6 micromachines-14-01165-f006:**
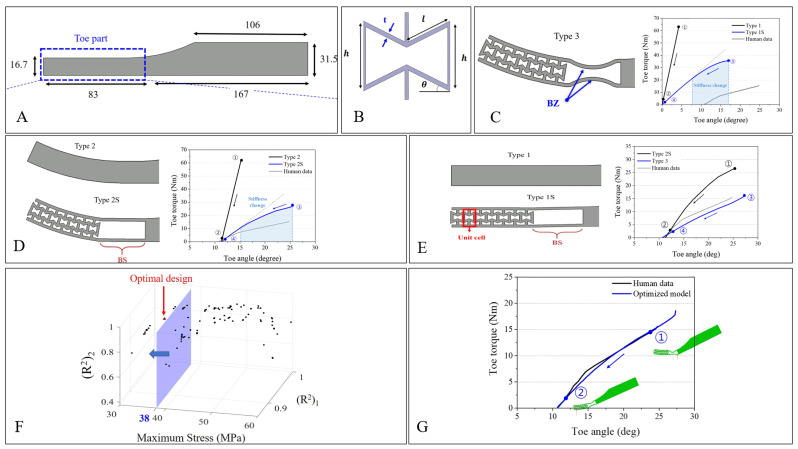
The use of auxetic materials in prosthetics for the toe part of the foot. (**A**) The overall dimension of the prosthetic foot, with the indicated area denoting the location for implementing the auxetic structure. (**B**) The design parameter used for the re-entrant hexagon used in this study. (**C**–**E**) The modelled foot variance from Type 1, 1S, 2, 2S, and 3 (left side of each figure); the right side shows the results of toe torque over the toe angle from the FEM. (**F**) Visualization of the end process of the optimization method using NSGA-II to gain the optimal design. For validation, this optimized design was then compared to the human gait data. From the graph shown (**G**), it is concluded that the optimized design shows a good agreement in mimicking the behaviour of the natural human toe [33].

**Figure 7 micromachines-14-01165-f007:**
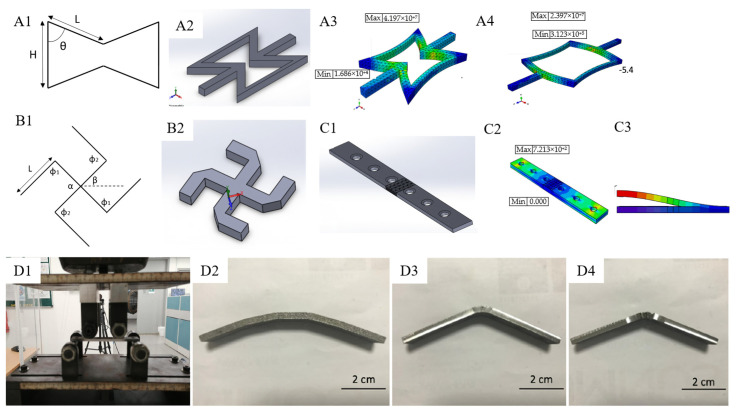
The use of auxetic materials in prosthetics for an orthopaedic bone plate. (**A1**,**B1**) The design parameters considered for each type of structure (re-entrant hexagon and missing rib, respectively). (**A2**,**B2**) The 3D-modelled CAD produced using SolidWorks. (**A3**,**A4**) The results for preliminary FEA to evaluate the behaviour of the unit cell when in tensile of 20% strain ((**A3**) for IA = 50 degrees, and (**A4**) for IA = 90 degree). (**C1**) The CAD model for the bone plate specimen with the positioning of the auxetic structure in the middle. (**C2,C3**) Visualization of the results of finite element analysis in terms of (**C2**) stress contours and (**C3**) displacement contours. (**D1**) An equivalent experimental test was conducted based on ASTM F-32 on the 3D-printed stainless steel specimens using an INSTRON machine. (**D2**–**D4**) Photographs for the specimen after the experimental test for the control specimen, re-entrant hexagon, and missing rib structure, respectively [39].

**Figure 8 micromachines-14-01165-f008:**
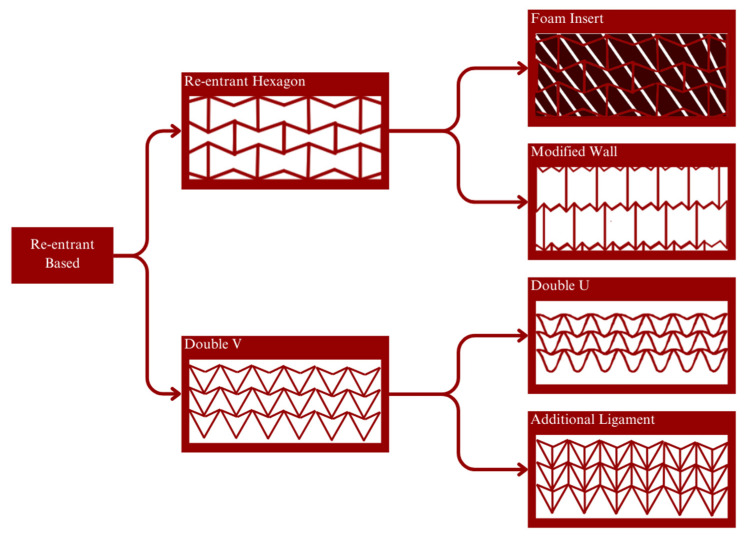
Development and improvement of re-entrant structure: Re-entrant structures have potential applications in various fields, including aerospace, biomedical engineering, and energy storage. They offer a promising avenue for creating lightweight, high-strength materials with unique mechanical properties. As research in this area continues to progress, we may see even more significant advancements and innovations in the design and development of re-entrant structures. Re-entrant structures are materials or structures that have repeated branching and folding patterns at different scales. They can be found in many natural materials, such as collagen fibres, bones, and plant tissues. Recently, re-entrant structures have gained significant attention in materials science and engineering due to their unique mechanical properties, including a high stiffness, strength, and toughness.

**Figure 9 micromachines-14-01165-f009:**
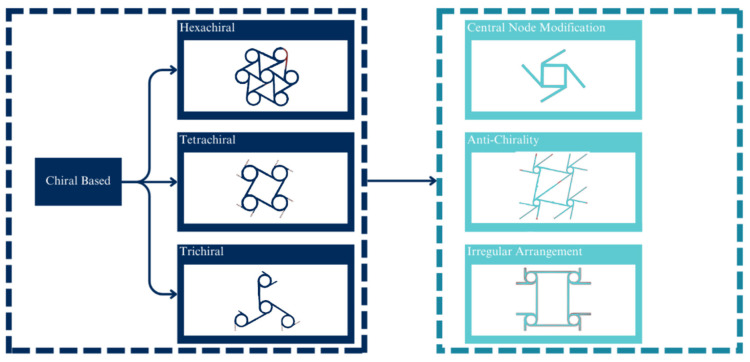
Development and improvement of chiral structure: Chiral structures also have potential applications in various fields, including optics, sensing, and energy storage. They offer a promising avenue for creating materials with unique properties that can be tailored for specific applications. As research in this area continues to progress, we may see even more significant advancements and innovations in the design and development of chiral structures. Chiral structures are structures that have mirror-image symmetry, meaning that they cannot be superimposed on their own reflection. They can be found in various natural materials, such as spider silk and collagen fibres. Chiral structures have gained significant attention in recent years due to their unique mechanical, optical, and biological properties.

**Figure 10 micromachines-14-01165-f010:**
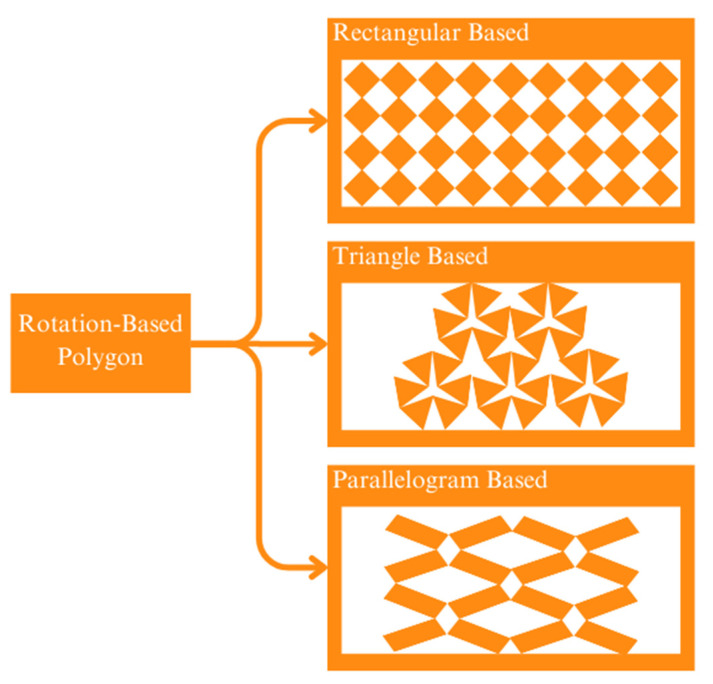
Subclasses of rotation-based polygon: Rotation-based polygons are a type of self-intersecting polygon that are formed by repeatedly rotating a line segment around a fixed point. These polygons can be classified into several subclasses based on their geometric properties. One subclass of rotation-based polygons is the monotone subclass, which is a polygon that can be divided into two monotone chains such that each chain is non-intersecting and has the property that any horizontal line intersects the chain at most twice. Monotone polygons have useful properties that mean they are well-suited for certain algorithms, such as triangulation algorithms. Another subclass of rotation-based polygons is the star subclass, which is a polygon that has a single vertex from which all other vertices are visible. Star polygons can be further classified into convex and non-convex subclasses. Convex star polygons are those in which all internal angles are less than 180 degrees, while non-convex star polygons have at least one internal angle greater than 180 degrees.

**Figure 11 micromachines-14-01165-f011:**
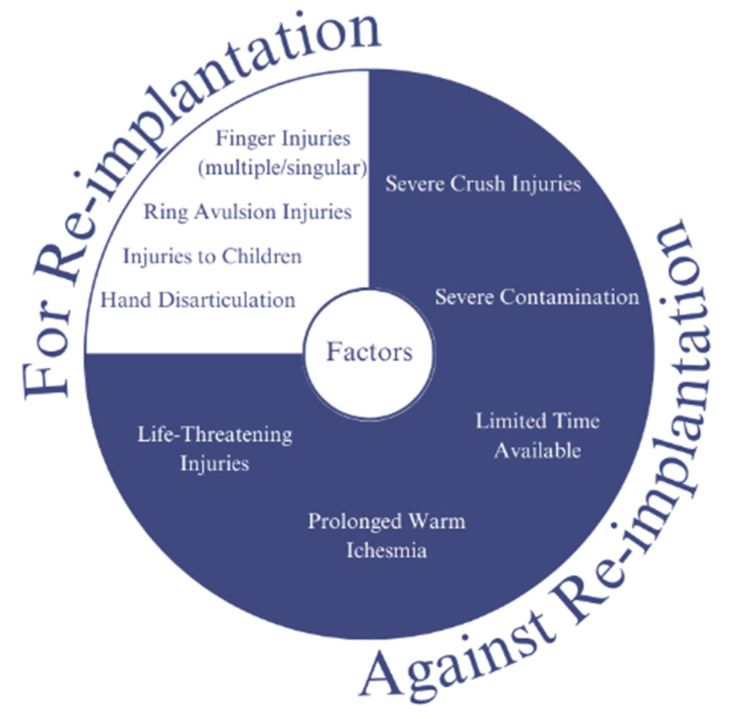
Consideration of replantation for post-traumatic disarticulation: Replantation is a surgical procedure that involves reattaching a body part, such as a finger or limb, that has been completely severed from the body. Post-traumatic disarticulation refers to the loss of a limb or body part as a result of a traumatic injury. In cases where replantation is not possible or not recommended, other treatment options, such as prosthetics or physical therapy, may be considered to help patients regain function and improve their quality of life. Ultimately, the decision to pursue replantation must be made on a case-by-case basis, with careful consideration of the patient’s individual circumstances and needs.

**Figure 12 micromachines-14-01165-f012:**
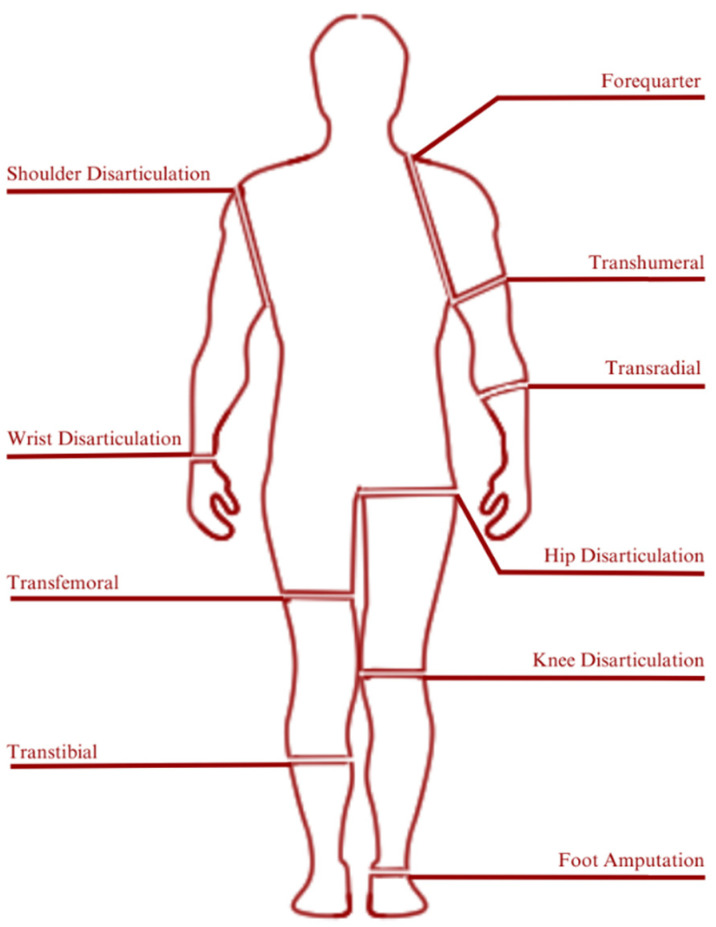
Prosthetic classification based on amputation/disarticulation levels: Prosthetics are artificial devices that replace missing body parts or limbs. Prosthetic limbs are designed to compensate for the loss of a limb, and the type of prosthetic required often depends on the level of amputation or disarticulation. The level of amputation or disarticulation determines the type of prosthetic limb that is required, and the prosthetic must be carefully fitted and adjusted to the individual’s needs to ensure the best possible function and comfort. Of the existing prosthetic classifications, transtibial is under discussion in this review. The transtibial prosthetic has several parts. Of the several existing parts, metamaterials are used in the development of prosthetic technology. These include parts of the sole, socket, foot, and suspension parts.

**Figure 13 micromachines-14-01165-f013:**
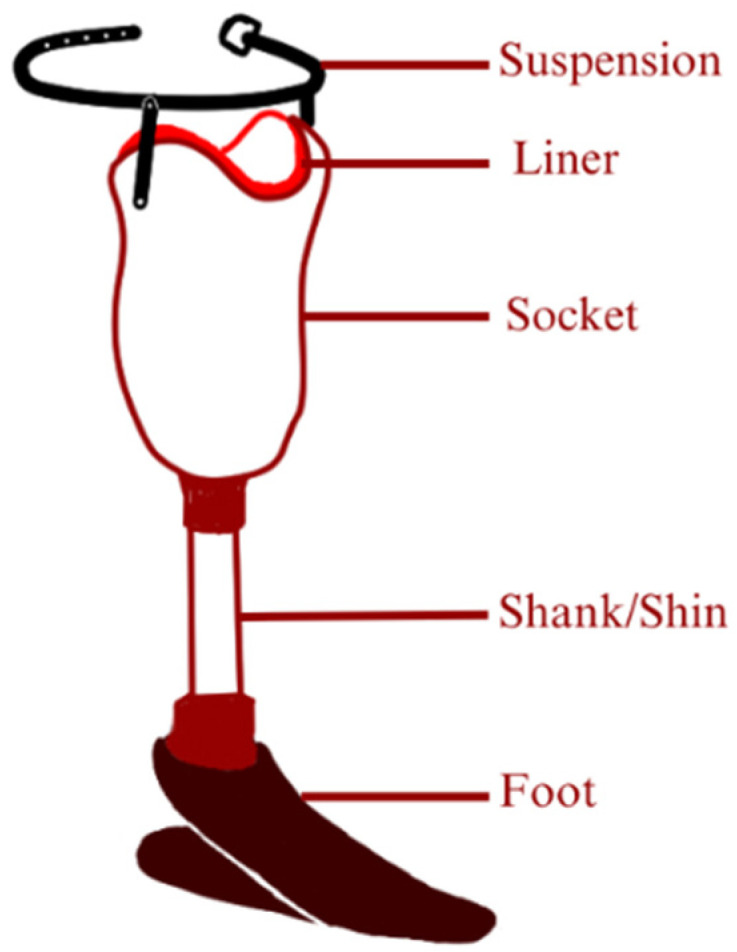
Composition of transtibial prosthetic: It is possible to develop the components of a transtibial prosthetic using auxetic metamaterials, but it would depend on the specific application and requirements of the prosthetic. Auxetic materials are unique in that they have a negative Poisson’s ratio, meaning that they expand in all directions when stretched. This property can be advantageous for certain applications, such as providing better shock absorption or improving energy return. For example, auxetic materials have been studied for use in prosthetic sockets to provide improved comfort and stability for amputees. The unique expansion properties of the auxetic material can help distribute pressure more evenly over the residual limb, reducing discomfort and the risk of injury. However, it is important to note that the design and development of prosthetic components using auxetic metamaterials is still a relatively new area of research, and there are many factors to consider when selecting materials and designing components for prosthetics. These include factors such as strength, durability, weight, and cost, among others.

**Figure 14 micromachines-14-01165-f014:**
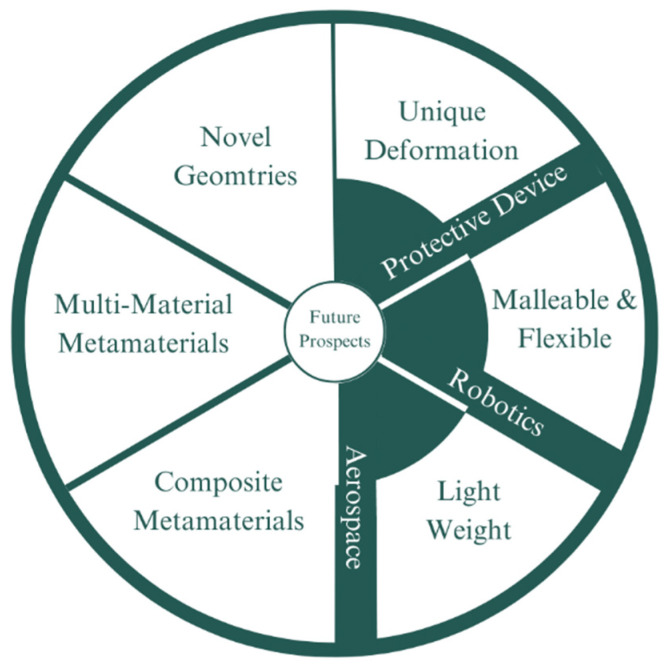
Potential and future prospects of auxetic metamaterials: Auxetic metamaterials have the potential to revolutionize the design and function of prosthetic transtibial components in several ways. Here, we name several potential and future prospects of auxetic metamaterials for transtibial prosthetics. Improved comfort: Auxetic materials have been shown to possess unique mechanical properties that can provide improved comfort for amputees. For example, auxetic materials can distribute pressure more evenly over the residual limb, reducing discomfort and the risk of injury. Better shock absorption: The expansion properties of auxetic materials can help absorb shocks and vibrations, which can be beneficial for amputees who engage in physical activities such as running or jumping. Customizable designs: Auxetic metamaterials can be engineered to exhibit specific properties and shapes, making them highly customizable for individual needs. This can result in more personalized and effective prosthetic transtibial components. Reduced weight: Many auxetic materials are lightweight and can be used to reduce the weight of prosthetic transtibial components, which can improve mobility and reduce fatigue. Durability: Certain auxetic materials have been shown to possess improved durability compared to traditional materials, which can lead to longer-lasting prosthetic components. Energy return: Certain auxetic materials can store and release energy, which can be beneficial for amputees who require more energy-efficient prosthetic transtibial components. Overall, the future prospects of auxetic metamaterials for transtibial prosthetics are promising. However, further research and development are needed to optimize the design and properties of these materials for specific applications in prosthetics. Additionally, cost-effectiveness and availability are important considerations when developing new materials and technologies for prosthetics.

**Table 1 micromachines-14-01165-t001:** Poisson’s ratio of common materials.

Materials	Poisson’s Ratio
Stainless Steel [45]	0.2535–0.2774
Thermoplastic Polyurethane Foam [46]	0.25
Nanoporous Gold [47]	0.4
Carbon Fibre [48]	0.26–0.28
Cork [49,50]	0
Cat Skin [51,52]	−0.3

**Table 2 micromachines-14-01165-t002:** Methodological comparison of past studies.

Name/Title	Objective	Comparison Aspect			
		FEA Tool	FEA Method	Material	Validation
General Comparison [82]	Comparing mechanical properties of past Auxetic Geometries	NX-Nastran	Compression	Epoxy Resin	Past Data
3D Re-entrant Hexagon [83]	Developing analytical model	Solidworks COSMOS	Compression	VeroWhitePlus	Experimental
Ancient Motif [84]	Developing structure based on existing (ancient) geometries	N/A	Tensile	Natural Latex Rubber	Experimental
Blast resistance (re-entrant hexagon) [22]	Investigating blast resistance of an auxetic panel	LS-DYNA	Blast Test	Aluminium Alloy	Experimental
Graded auxetic hexagon [85]	Investigating flexural properties of auxetic panel	ABAQUS	3-P-Flexural	PLA	Experimental DIC
Planar 3D chiral with rectangular central node [76]	Investigating mechanical properties of novel arrangement for 3D Chiral	ANSYS APDL	Compression	UV Curable Resin	Experimental
Shape matching [86]	Developing the concept of shape-matching	ABAQUS	Tensile	PLA	Experimental
Non-positive thermal expansion [87]	Developing 3D structure with two unique behavior	ANSYS	Compression	Steel-Invar and Aluminium-Invar	Numerical
Star honeycomb [88]	Investigating crushing behavior on star honeycomb	LS-DYNA	Crushing	Aluminium Alloy	Numerical
Peanut inspired [89]	Developing 2D structure based on natural geometries	ABAQUS	Tensile	PLA	Experimental
Turtle inspired [90]	Developing 2D structure based on natural geometries	ABAQUS	Compression	Aluminium	Numerical
4D-Printing SMP [91]	Developing Shape-Memory-Alloy	ANSYS	Tensile	SMP FlexPro	Experimental
Foam for structure [61]	Investigating the effect of filler foam in hexagonal structure	ABAQUS	Compression	TPU SR and FR Foam	Experimental
Ballistic resistance [92]	Investigating the potential of auxetic for ballistic resistance	ABAQUS	Ballistic Impact	Carbon Fiber Epoxy Resin	Experimental
Foam for tubular auxetic [62]	Investigating the effect of filler foam in tubular auxetic structure	ABAQUS	Compression	Stainless Steel PU Foam	Experimental
Additional node for re-entrant hexagon [67]	Modifying the design of re-entrant hexagon by applying additional nodes	ABAQUS	Compression	ABS	Experimental
Stretching dominated deformation [93]	Developing a structure with deformation behavior that is dominated by stretching	ABAQUS	Compression	CFRP	Experimental
Double U [19]	Improving mechanical properties by converting into curve (Double U)	ABAQUS	Compression	Stainless Steel	Experimental
Additional ligament DAH and re-entrant hexagon [69]	Improving stiffness by adding ligament	ABAQUS	Tensile	SLA	Experimental
3D-Planar anti-chiral [94]	Implementation of oblique node on auxetic structure	ABAQUS	Tensile	VeroWhitePlus	Experimental
Graded chiral [95]	Investigating the out-of-plane impact energy absorption of graded chiral	ABAQUS	Dynamic Crushing	DP590 Steel	Numerical
Auxetic stent [24]	Designing auxetic stent for CAD	ABAQUS	Practical Simulation	316L Stainless Steel	Theoretical
Ballistic resistance honeycomb sandwich [96]	Examining the performance of HSP with auxetic structure	ANSYS and LS DYNA	Ballistic impact simulation	Aluminium alloy AA6060 T4	Theoretical
Inverted tetrapod [97]	Proposing the base geometry of inverted tetrapod as auxetic structure	LS-DYNA	Quasi-static	Ti-6A1-4V Alloy powder	Experimental
Out-of-plane ballistic performance [98]	Exploring the performance of out-of-plane ballistic performance of different HSP	ABAQUS	Ballistic impact simulation	5052-H39 Aluminium sheets	Numerical
RPC filler for honeycomb [99]	Examining the performance of auxetic HSP filled with RPC	LS-DYNA	Ballistic impact simulation	45 Steel	Numerical
Sandwich panel with CFRP sheet [100]	Applying a CFRP as face sheet for auxetic HSP	LS-DYNA	Ballistic impact	AlSi10Mg	Experimental
Auxetic in doubly curved HSP [101]	Implementation of oblique node on auxetic structure	ABAQUS	Tensile	VeroWhitePlus	Experimental
Modified re-entrant honeycomb [21]	Additional horizontal member between vertical and re-entrant on a semi-re-entrant honeycomb model	Soliworks and ABAQUS	Tensile	Acrylic Sheet	Experimental and numerical

**Table 3 micromachines-14-01165-t003:** Limitations and comparison for state-of-the-art prosthetics.

Comparison	State-of-the-Arts				
	Transtibial Socket Inlay	Transfemoral Socket Liner	Heel-Off Foot	Toe-Off Foot	Bone Implant
Testing Method	Both	FEA	FEA	FEA	Both
Sample Material Fabrication	Yes	No	No	No	Yes
Prototyping	No	No	No	No	Yes
Sample Experimental Testing	ASTM D575	No	No	No	ASTM F-32
Prototype Practical Testing	No	No	No	No	No
Validation	Experimental	Numerical	Numerical	Gait Data	Experimental

## Data Availability

Not applicable.

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
