# Peer review of "Revolutionizing Prosthetic Design with Auxetic Metamaterials and Structures: A Review of Mechanical Properties and Limitations"

_micromachines, 2023, doi:10.3390/mi14061165_

Round 1

Reviewer 1 Report

Manuscript Notes: Revolutionizing Prosthetic Design with Auxetic Metamaterials: A Review of Mechanical Properties and Limitations

 The work submitted for evaluation impresses with the extensiveness of the collected publication material. These works present current research trends in the field of mechanical metamaterials, which are based on the simulation of the behavior of idealized models of auxetic structures and with the use of computer graphics. This approach is a kind of sailing in the virtual world, but the conclusions are very interesting. For real auxetic structures, many technical problems need to be solved, and without them, prosthetics cannot use auxetics, in particular, it is about the implementation and selection of materials for connections between unit cells in the construction of an auxetic structure.

It can also be added that mechanical metamaterials are used in prosthetics, which are not necessarily auxetics.

I must recall some important properties of the auxetic constructions known so far.

Real auxetic materials show very little deformation under load. After all, they concern the behavior of flexible materials forming connections of rigid unit cells, whether in the form of rotating figures, or forming connections of struts in re-entrant cells, or of the arrowhead or star type. The stiffness of the unit cells is necessary so that the applied stress can transfer the unit cells to each other. Auxetic structures are unit cells joined together by an elastic material. It is also known that most elastic-elastic materials have a strain of less than 0.2. Connections between unit cells in auxetic structures undergo bending and elongation or contraction under load, and only to a small extent. Simulations, on the other hand, show strain from 0 to 1! After all, the percentage expansion of real constructions of auxetic structures subjected to stretching does not exceed a few percent, and at higher values the structures are destroyed. It also seems that elongation or contraction under load in auxetic structures is more important for prostheses than Poisson's ratio.

Considering the above, one has to be careful in assessing the value of these usually elegant and often based on higher mathematics simulation studies - also in this case when discussing the simulation of auxetic structures predicted as prostheses or prosthesis components. Nevertheless, these simulations may point the way for further development of prosthetics using mechanical metamaterials. From this point of view, this manuscript should be assessed positively.

At the end of my opinion, I would like to point out a few minor shortcomings. Among the metamaterials, the most important are optical metamaterials (Vasylago, Pendry), which the authors did not mention next to mechanical, acoustic, electromagnetic and thermal metamaterials. Figures, if not created by the authors, must be cited.

Author Response

Dear Reviewer 1 We have carefully revised the manuscript based on your inputs. Kindly find our response in the attached file. Sincerely

Reviewer 2 Report

This is a nicely written review which would be of interest for readers. It is not very scientific, there is nothing much to review on it. The authors shall carefully read the text again  to correct minor mistakes in English grammar. There are many on every page of the manuscript - few examples are suggested by comments below

The referee suggest a change of title

 Revolutionizing Prosthetic Design with Auxetic Metamaterials and Structures: 2 A Review of Mechanical Properties and Limitations

because many materials reviewed  in the manuscript are actually structures

Comments

P1    Infection, disease (related to the circulatory system), and traumatic 38 events (falling, crushed, blasted)…. This is not sentence

P3  Lakes first studied this unique 90 property in 1987 [10], …. Lakes at al. [10] first studied this unique 9property in 1987,

P3  In this review, we provide recent state-of-the-art research in the field of prosthetics 106 and discuss….we provide an overview on recent…

Etc.

language is almost fine, the text shall be just a bit polished once more

Author Response

Dear Reviewer 2 We have carefully revised the manuscript based on your inputs. Kindly find our response in the attached file. Sincerely

Reviewer 3 Report

The manuscript “Revolutionizing Prosthetic Design with Auxetic Metamaterials: A Review of Mechanical Properties and Limitations” by Fardan et al. reviews the current state of the art in the development of prosthetics using auxetic metamaterials. Although the originality of some figures is unclear, this manuscript is well written. Therefore, I would suggest authors may take a revision before publication. Here are the comments and suggestions:

1.     The originality of some figures is unclear, please add citations if necessary.

Author Response

Dear Reviewer 3 We have carefully revised the manuscript based on your inputs. Kindly find our response in the attached file. Sincerely
